# CONSISTENT4D: CONSISTENT 360° DYNAMIC OBJECT GENERATION FROM MONOCULAR VIDEO

**Yanqin Jiang**[1,2], **Li Zhang**[3], **Jin Gao**[1,2], **Weiming Hu**[1,2,6], **Yao Yao**[4,5] ✉

[1]State Key Laboratory of Multimodal Artificial Intelligence Systems (MAIS), CASIA
[2]School of Artificial Intelligence, University of Chinese Academy of Sciences
[3]School of Data Science, Fudan University
[4]State Key Laboratory for Novel Software Technology, Nanjing University
[5]School of Intelligence Science and Technology, Nanjing University
[6]School of Information Science and Technology, ShanghaiTech University
`jiangyanqin2021@ia.ac.cn, lizhangfd@fudan.edu.cn,`
`{jin.gao, wmhu}@nlpr.ia.ac.cn, yaoyao@nju.edu.cn`

## ABSTRACT

In this paper, we present Consistent4D, a novel approach for generating 4D dynamic objects from uncalibrated monocular videos. Uniquely, we cast the 360-degree dynamic object reconstruction as a 4D generation problem, eliminating the need for tedious multi-view data collection and camera calibration. This is achieved by leveraging the object-level 3D-aware image diffusion model as a supervision signal for training dynamic Neural Radiance Fields (DyNeRF). Specifically, we propose a cascade DyNeRF to facilitate stable training convergence and temporal continuity given the time-discrete supervision signal. To achieve spatial and temporal consistency of the 4D generation, an interpolation-driven consistency loss is further introduced, which aligns the rendered frames with the interpolated frames from a pre-trained video interpolation model. Extensive experiments show that the proposed Consistent4D significantly outperforms previous 4D reconstruction approaches as well as per-frame 3D generation approaches, opening up new possibilities for 4D dynamic object generation from a single-view uncalibrated video. Project page: https://consistent4d.github.io.

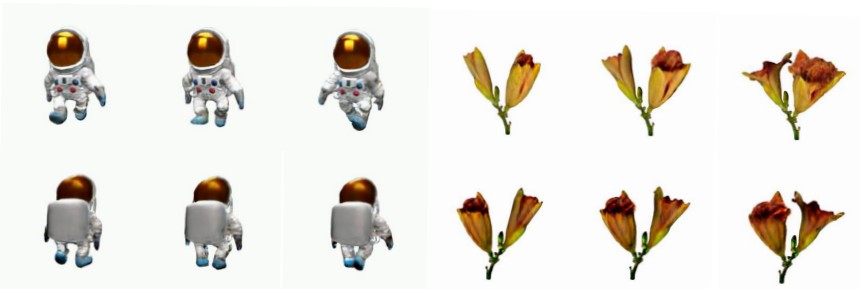

Figure 1: Video-to-4D generation results of two input videos. We show the renderings at 2 viewpoints and 3 timestamps to demonstrate the spatial-temporal consistency achieved by Consistent4D.

## 1 INTRODUCTION

Dynamic 3D content creation, is critical for a variety of downstream applications, including virtual reality, augmented reality, autonomous driving simulation, and gaming. Existing 4D reconstruction methods typically rely on synchronized multi-view videos (Li et al., 2022; Fridovich-Keil et al., 2023; Shao et al., 2023) or calibrated monocular videos (Li et al., 2021; Pumarola et al., 2021; Park et al., 2021b;a). These approaches, although effective, necessitate complex camera setups and rigorous camera calibrations, constraining their usability and scalability in real-world settings. They also struggle with reconstructing unobserved regions.

To adresss the aforementioned limitations, we instead generate 360° dynamic 3D object from a single-view uncalibrated video captured by a static camera, streamlining data acquisition while maintaining high fidelity outputs. To tackle the challenging problem of 4D scene recovery from a single video, we harness visual priors from pre-trained large vision models (LVMs), casting 4D reconstruction as a generative process. This is inspired by human cognitive abilities, which intuitively interpret 3D shape, appearance, and motion from a video clip using knowledge accumulated through life.

Inspired by recent advancements in 3D generation techniques (Poole et al., 2023; Wang et al., 2022; Chen et al., 2023; Wang et al., 2023; Deng et al., 2022; Tang et al., 2023; Melas-Kyriazi et al., 2023), we introduce multi-view diffusion model (Liu et al., 2023) to provide spatially consistent supervision signal to the generation of dynamic 3D object. However, this supervision signal is time-discrete, posing challenges for maintaining temporal coherence in dynamic 3D object generation and consequently compromising the spatiotemporal consistency in overall 4D optimization.

In this work, we present Consistent4D, a novel video-to-4D generation approach with focus on spatial and temporal consistency. Our approach features a Cascade DyNeRF tailored for 4D scene representation in generation tasks, which facilitates temporal coherence under time-discrete supervison singal from a pre-trained multi-view diffusion model. To further improve both spatial and temporal consistency, an Interpolation-driven Consistency Loss (ICL) is introduced, which minimizes the discrepancy between frames rendered by DyNeRF and frames interpolated by a pre-trained video interpolation model. The ICL loss not only enhances consistency in 4D generation but also mitigates multi-face issues in 3D generation. Finally, an optional video enhancer refines occasional noise in 4D renderings.

We have extensively evaluated our approach on both synthetic videos rendered from animated 3D models and in-the-wild videos collected from the Internet. To summarize, contributions of this work include:

- We propose a video-to-4D framework for dynamic object generation from a statically captured monocular video. A Cascade DyNeRF, inherently inclined to temporally consistent outputs, is optimized by a multi-view image diffusion model, with the option to refine its renderings using a video enhancer.
- We introduce a novel Interpolation-driven consistency loss to improve the spatial and temporal consistency of the 4D generation, which also alleviates multi-face Janus problem in 3D generation.
- We extensively evaluate our method on both synthetic and in-the-wild videos collected from the Internet, showing promising results for the new task of video-to-4D generation.

## 2 RELATED WORK

**3D Generation** Recently, general-purpose 3D generation has been enabled by text-to-image diffusion models pre-trained on Internet-scale data. DreamFusion (Poole et al., 2023) stands for the pioneering work of text-to-3D by using a 2D diffusion model, where the Score Distillation Sampling (SDS) loss is proposed to leverage the denoise diffusion process for Neural Radiance Field (NeRF) training. The follow-up works (Lin et al., 2023; Chen et al., 2023; Wang et al., 2023) further enhance the visual quality of the generated object by using mesh representation, Variational Score Distillation(VSD), etc.

Alongside text-to-3D, image-to-3D is a popular alternative for 3D generation. Different from 3D reconstruction, which focuses on recovering 3D information from overlapped multi-view images, image-to-3D generation usually takes only one single image as input and relies on data priors (e.g., 2D diffusion) to generate invisible regions of the object (Melas-Kyriazi et al., 2023; Tang et al., 2023; Liu et al., 2023). Most approaches in this domain simply convert input image to texts by using a large vision-language models and then reused the text-to-3D model for image-to-3D generation (Melas-Kyriazi et al., 2023; Tang et al., 2023; Seo et al., 2023). One of the biggest challenges in 3D generation is multi-face Janus problem, arising from the lack of 3D awareness in the original 2D diffusion model. However, the recent image-to-3D work Zero123 (Liu et al., 2023) mitigates this issue by fine-tuning a 3D-aware image-to-image model on a large-scale multi-view dataset, enabling the novel view synthesis of the object from the input image. Our work employs Zero123

and we propose an Interpolation-driven consistency Loss to further enhance spatial and temporal consistency.

**4D Reconstruction** Early works in 4D reconstruction, aka dynamic 3D reconstruction, are mainly object-level and adopt parametric shape models (Matthew Loper & Black, 2015; Vo et al., 2020) as representation. In recent years, Dynamic Neural Radiance Field(DyNeRF) become popular, and convenient dynamic scene reconstruction is enabled. These works can be classified into two categories: a deformed scene is directly modeled as a NeRF in canonical space with a time-dependent deformation (Pumarola et al., 2021; Park et al., 2021a;b; Wu et al., 2022b; Tretschk et al., 2021) or time-varying NeRF in the world space (Gao et al., 2021; Li et al., 2021; Xian et al., 2021; Fridovich-Keil et al., 2023; Cao & Johnson, 2023). These methods usually require multi-view, synchronized, and calibrated videos as input to the reconstruction algorithms, which is tedious to collect Li et al. (2022); Shao et al. (2023). In other scenarios, a monocular video is accepted as input, but still requires accurate per-frame calibration and a moving trajectory for effective dynamic information recovery (Pumarola et al., 2021; Park et al., 2021a;b). In contrast, our approach utilizes a single-view, uncalibrated video as input, significantly streamlining the data collection.

**4D Generation** 4D generation extends 3D generation to the space+time domain and thus is more challenging. Early attempts focus on category-specific generation and parametric shape models is employed as scene representation (Zuffi et al., 2017; 2018; Vo et al., 2020; Kocabas et al., 2020) . They usually take images or videos as conditions and need category-specific 3D templates or per-category training from a collection of images or videos (Ren et al., 2021; Wu et al., 2021; Yang et al., 2022; Wu et al., 2022a). Recently, MAV3D (Singer et al., 2023) has achieved general-purpose dynamic scene generation based on textual descriptions. It follows the paradigm of Dream-Fusion (Poole et al., 2023) and extends it to the time domain by proposing a three-stage training strategy. However, the quality of generated scenes is limited due to low-quality video diffusion models. Instead, we propose the task of video-to-4D generation, leveraging the high-quality input video as a guiding constraint to enhance fidelity and ensure consistent rendering in the 4D output.

## 3 PRELIMINARIES

### 3.1 SCORE DISTILLATION SAMPLING FOR IMAGE-TO-3D

Score Distillation Sampling (SDS) is first proposed in DreamFusion (Poole et al., 2023) for text-to-3D task. It enables the use of a 2D text-to-image diffusion model as a prior for optimization of a NeRF. We denote the NeRF parameters as $\theta$, text-to-image diffusion model as $\phi$, text prompt as $\rho$, the rendering image and the noisy image as $\mathbf{x}$ and $\mathbf{z}$, the SDS loss is defined as:

$$\nabla_\theta \mathcal{L}_{SDS}(\phi, \mathbf{x}) = \mathbb{E}_{\tau, \epsilon} \left[ \omega(t) \left( \hat{\epsilon}_\theta(\mathbf{z}_t; \rho, \tau) - \epsilon \right) \frac{\partial \mathbf{x}}{\partial \theta} \right], \quad (1)$$

where $\tau$ is timestamps in diffusion process, $\epsilon$ denotes noise, and $\omega$ is a weighted function. Intuitively, this loss perturbs $\mathbf{x}$ with a random amount of noise corresponding to the timestep $\tau$, and estimates an update direction that follows the score function of the diffusion model to move to a higher density region.

Besides text-to-3D, SDS is also widely used in image-to-3D tasks. Zero123 (Liu et al., 2023) is one prominent representative. It proposes a viewpoint-conditioned image-to-image translation diffusion model fine-tuned from Stable Diffusion (Rombach et al., 2022), and exploits this 3D-aware image diffusion model to optimize a NeRF using SDS loss. This image diffusion model takes one image, denoted by $\mathbf{I}_{in}$, and relative camera extrinsic between target view and input view, denoted by $(\mathbf{R}, \mathbf{T})$, as the input, and outputs the target view image $\mathbf{I}_{out}$. Compared with the original text-to-image diffusion model, text prompt in Equation 1 is not required in this model cause the CLIP embedding of the input image and the relative viewpoint change replace the text prompt. Then Equation 1 could be re-written as:

$$\nabla_\theta \mathcal{L}_{SDS}(\phi, \mathbf{x}) = \mathbb{E}_{\tau, \epsilon} \left[ \omega(t) \left( \hat{\epsilon}_\theta(\mathbf{z}_t; \mathbf{I}_{in}, \mathbf{R}, \mathbf{T}, \tau) - \epsilon \right) \frac{\partial \mathbf{x}}{\partial \theta} \right], \quad (2)$$

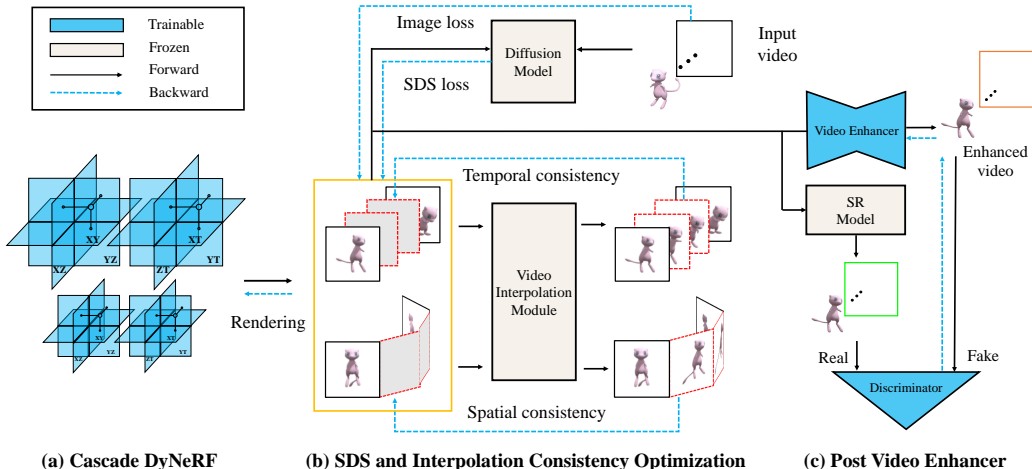

Figure 2: Schematic illustration of Consistent4D. We introduce Cascade DyNeRF as our 4D representation, specially tailored to favor temporal continuity, ensuring stable training under time-discrete SDS from a multi-view diffusion model. To compensate for temporal inconsistency arising from SDS, we propose a novel Interpolation Consistency Loss, leveraging a pre-trained video interpolation model, to enhance both spatial and temporal consistency of 4D outputs. Additionally, we offer an optional video enhancer as post-processing step to refine any potential noise in 4D renderings.

## 3.2 K-PLANES

K-planes (Fridovich-Keil et al., 2023) is a simple and effective dynamic NeRF method which factorizes a dynamic 3D volume into six feature planes (i.e., hex-plane), denoted as $P = \{P_o\}$, where $o \in \{xy, yz, xz, xt, yt, zt\}$. The first three planes correspond to spatial dimensions, while the last three planes capture spatiotemporal variations. Each of the planes is structured as a $M \times M \times F$ tensor in the memory, where $M$ represents the size of the plane and $F$ is the feature size that encodes scene density and color information. Let $t$ denote the timestamp of a video clip, given a point $p = (x, y, z, t)$ in the 4D space, we normalize the coordinate to the range $[0, M]$ and subsequently project it onto the six planes using the equation $f(p)_o = P_o(\iota_o(p))$, where $\iota_o$ is the projection from a space point $p$ to a pixel on the $o$'th plane. The plane feature $f(p)_o$ is extracted via bilinear interpolation. The six plane features are combined using the Hadamard product (i.e., element-wise multiplication), to produce a final feature vector as follows:

$$f(p) = \prod_{o \in \{xy, yz, xz, xt, yt, zt\}} f(p)_o, \tag{3}$$

Then, the color and density of $p$ is calculated as $c(p) = c(f(p))$ and $d(p) = d(f(p))$, where $c$ and $d$ denotes mlps for color and density.

## 4 METHOD

In this work, we target to generate a 360° dynamic object from a statically captured monocular video. To achieve this goal, we develop a framework consisting of a DyNeRF and an optional video enhancer, supervised by the pre-trained multi-view diffusion model and a GAN, respectively. As shown in Figure 2, we design a Cascade DyNeRF which naturally favors temporal continuity, facilitating stable training under the time-discrete SDS loss from 2D diffusion model. To further guarantee spatial and temporal consistency, we propose a novel Interpolation-driven Consistency Loss as the extra regularization for the DyNeRF. Inspired by pix2pix (Isola et al., 2017), we apply GAN to train a lightweight video enhancer as an optional post processing step to refine the occasional noise in 4D renderings. In such a way, we obtain a DyNeRF from which we can render 360° view of the dynamic object, and the rendered results can be further enhanced by the video enhancer.

In the following sections, we will first introduce our design of the Cascade DyNeRF, and then illustrate the Interpolation-driven Consistency loss. Video enhancer is described in the third section. At last, we detail the training loss.

## 4.1 CASCADE DyNeRF

Existing DyNeRF methods mainly assume the supervision signals are temporally coherent, however, this assumption does not hold in our pipeline due to the use of the image diffusion models with no time notion. In order to minimize the impact of temporal discontinuity in the supervision signals, we are prone to 4D representations which naturally guarantee a certain level of temporal continuity. Therefore, we build our DyNeRF based on K-planes (Fridovich-Keil et al., 2023) which exploits temporal interpolation, an operator naturally inclined to temporal smoothing. Empirically, maintaining temporal coherence is possible when the time resolution of spatiotemporal planes is small, however, this results in over-smoothed images where finer details are lost. In contrast, when the time resolution is large, the quality of the images is enhanced, but the continuity of images within the same time series diminishes. To achieve both temporal continuity and high image quality, we adjust the multi-scale technique from the original paper and introduced Cascade DyNeRF.

Let us denote the scale index by $s$. In original K-planes, multi-scale features are exploited by concatenation along feature dimension, then the color and density could be calculated as:

$$c(p) = c(\text{concat}(\{f(p)^s\}_{s=1}^S)), \; d(p) = d(\text{concat}(\{f(p)^s\}_{s=1}^S)), \tag{4}$$

where $S$ is the number of scales. In our setting, simple concatenation is hard to balance between image quality and temporal consistency. So we propose to leverage the cascade architecture and let the low-resolution planes learn temporally coherent dynamic objects with a certain degree of over-smoothing, and let the high-resolution planes learn the residual between the above results and the target ones. The final color and density are the addition of results from planes across the scale. That is,

$$c(p)^s = \sum_{k=1}^s c(f(p)^k), \; d(p)^s = \sum_{k=1}^s d(f(p)^k), \tag{5}$$

where k indicates the scale index. Note that SDS loss and other losses are applied to the rendering results of each scale to guarantee that planes with higher resolution only learn the residual between results from previous scales and the target object. In this way, we can improve temporal consistency without sacrificing much object quality.

## 4.2 INTERPOLATION-DRIVEN CONSISTENCY LOSS

Video generation methods usually train an inter-frame interpolationi module to enhance the temporal consistency between keyframes (Ho et al., 2022; Zhou et al., 2022; Blattmann et al., 2023). Inspired by this, we exploit a pre-trained light-weighted video interpolation model and propose an Interpolation-driven Consistency Loss to enhance the spatiotemporal consistency of the 4D generation.

The interpolation model adopted in this work is RIFE (Huang et al., 2022), which takes a pair of consecutive images as well as the interpolation ratio $\gamma$ ($0 < \gamma < 1$) as the input, and outputs the interpolated image. In our case, we first render a batch of images that are either spatially continuous or temporally continuous, denoted by $\{\mathbf{x}\}_{j=1}^J$, where $J$ is the number of images in a batch. Let us denote the video interpolation model as $\psi$, the interpolated image as $\hat{\mathbf{x}}$, then we calculate the Interpolation-driven Consistency Loss as:

$$\hat{\mathbf{x}}_j = \psi(\mathbf{x}_1, \mathbf{x}_J, \gamma_j),$$
$$\mathcal{L}_{ICL} = \sum_{j=2}^{J-1} \|\mathbf{x}_j - \hat{\mathbf{x}}_j\|_2, \tag{6}$$

where $\gamma_j = \frac{j-1}{J-1}$, and $2 \le j \le J - 1$.

This simple yet effective loss enhances the continuity between frames thus improving the spatiotemporal consistency in dynamic object generation by a large margin. Moreover, we find the spatial version of this loss alleviates the multi-face problem in 3D generation tasks as well. Please refer to the

experiment sections to see quantitative and qualitative results. The Interpolation-driven Consistency Loss and some other regularization losses are added with SDS loss in Equation 2, details of which can be found in the experiment section.

### 4.3 Cross-frame Video Enhancer

Sometimes image sequence rendered from the optimized DyNeRF suffers from artifacts, such as blurry edges, small floaters, and insufficient smoothness, especially when the object motion is abrupt or complex. To further improve the quality of rendered videos, we design a lightweight video enhancer and optimize it via GAN, following pix2pix (Isola et al., 2017). The real images are obtained with image-to-image technique (Meng et al., 2021) using a super-resolution diffusion model, and the fake images are the rendered ones.

To better exploit video information, We add cross-frame attention to the UNet architecture in pix2pix, i.e., each frame will query information from two adjacent frames. We believe this could enable better consistency and image quality. Denote the feature map before and after cross-frame-attention as $\mathbf{F}$ and $\mathbf{F}'_j$, we have:

$$
\begin{aligned}
F'_j &= \text{Attention}(\mathcal{Q}_j, \mathcal{K}_j, \mathcal{V}_j), \\
\mathcal{Q}_j &= \text{flatten}(F_j), \mathcal{K}_j = \mathcal{V}_j = \text{flatten}(\text{concat}(F_{j-1}, F_{j+1})),
\end{aligned}
\tag{7}
$$

where $\mathcal{Q}$, $\mathcal{K}$ and $\mathcal{V}$ denotes query, key, and value in attention mechanism, and concat denotes the concatenation along the width dimension.

### 4.4 Opimization

We optimize the dynamic NeRF using SDS loss $\mathcal{L}_{SDS}$ in Eq. 2 and ICL loss $\mathcal{L}_{ICL}$ in Eq. 6. Besides, we apply reconstruction loss $\mathcal{L}_{rec}$ and foreground mask loss $\mathcal{L}_m$ for the input view following Guo et al. (2023). 3D normal smooth loss $\mathcal{L}_n$ (Guo et al., 2023) and orientation loss $\mathcal{L}_{ori}$ (Verbin et al., 2022) are utilized to achieve better geometry. Therefore, the final optimization objective for dynamic NeRF is calculated as:

$$
\mathcal{L} = \lambda_1 \mathcal{L}_{SDS} + \lambda_2 \mathcal{L}_{ICL} + \lambda_3 \mathcal{L}_{rec} + \lambda_4 \mathcal{L}_m + \lambda_5 \mathcal{L}_n + \lambda_6 \mathcal{L}_{ori}
\tag{8}
$$

For video enhancer in optional post-processing step, the loss function is the same as pix2pix (Isola et al., 2017).

## 5 Experiment

We have conducted extensive experiments to evaluate the proposed Consistent4D generator using both synthetic data and in-the-wild data. The experimental setup, comparison with dynamic NeRF baselines, and ablations are provided in the following sections.

### 5.1 Implementation Details

**Data Preparation** For qualitative experiments, we collect in-the-wild and synthetic videos from Internet. For quantitative evaluation, we select and download seven animated models, namely *Pistol*, *Guppie*, *Crocodie*, *Monster*, *Skull*, *Trump*, *Aurorus*, from Sketchfab (ske, 2023) and render the multi-view videos by ourselves, as shown in Figure 3 and appendix A.3. We render one input view for scene generation and 4 testing views for our evaluation. For each input video, we initially segment the foreground object utilizing SAM (Kirillov et al., 2023) and subsequently sample 32 frames uniformly. The majority of the input videos span approximately 2 seconds, with some variations extending to around 1 second or exceeding 5 seconds.

**Training** During SDS and interpolation consistency optimization, we utilize zero123-xl trained by Deitke et al. (2023) as the diffusion model for SDS loss. For Cascade DyNeRF, we set $s = 2$ in most experiments except for the last row in Table 1a, i.e., we have coarse-level and fine-level DyNeRFs. The spatial and temporal resolution of Cascade DyNeRF are configured to 50 and 8 for coarse-level, and 100 and 16 for fine-level, respectively. We first train DyNeRF with batch size 4 and resolution

| | image-level | | video-level | Cas-DyNeRF | ICL | Video enhancer | image-level | | video-level |
| --- | --- | --- | --- | --- | --- | --- | --- | --- | --- |
| | LPIPS ↓ | CLIP ↑ | FVD ↓ | | | | LPIPS ↓ | CLIP ↑ | FVD ↓ |
| D-NeRF | 0.51 | 0.68 | 2327.83 | | | | 0.16 | 0.86 | 1303.31 |
| K-planes | 0.38 | 0.72 | 2295.68 | ✓ | | | 0.16 | 0.87 | 1226.92 |
| Zero123 | 0.15 | 0.90 | 1571.60 | | ✓ | | **0.15** | **0.88** | 1205.80 |
| Ours (s=2) | 0.16 | 0.87 | **1133.44** | ✓ | ✓ | | 0.16 | 0.87 | **1133.44** |
| Ours (s=4) | **0.12** | **0.91** | **992.61** | ✓ | ✓ | ✓ | 0.16 | 0.87 | **1114.85** |

(a) Comparison with other methods.          (b) Ablations.

Table 1: Quantitative results on synthetic dataset.

64 for 5000 iterations. Then we decrease the batch size to 1 and increase the resolution to 256 for the next 5000 iteration training. ICL is employed in the initial 5000 iterations with a probability of 25%. The optimization of dynamic NeRF and video enhancer cost about 2.5 hours and 15 minutes on a single V100 GPU. For more details, please refer to the appendix A.5.

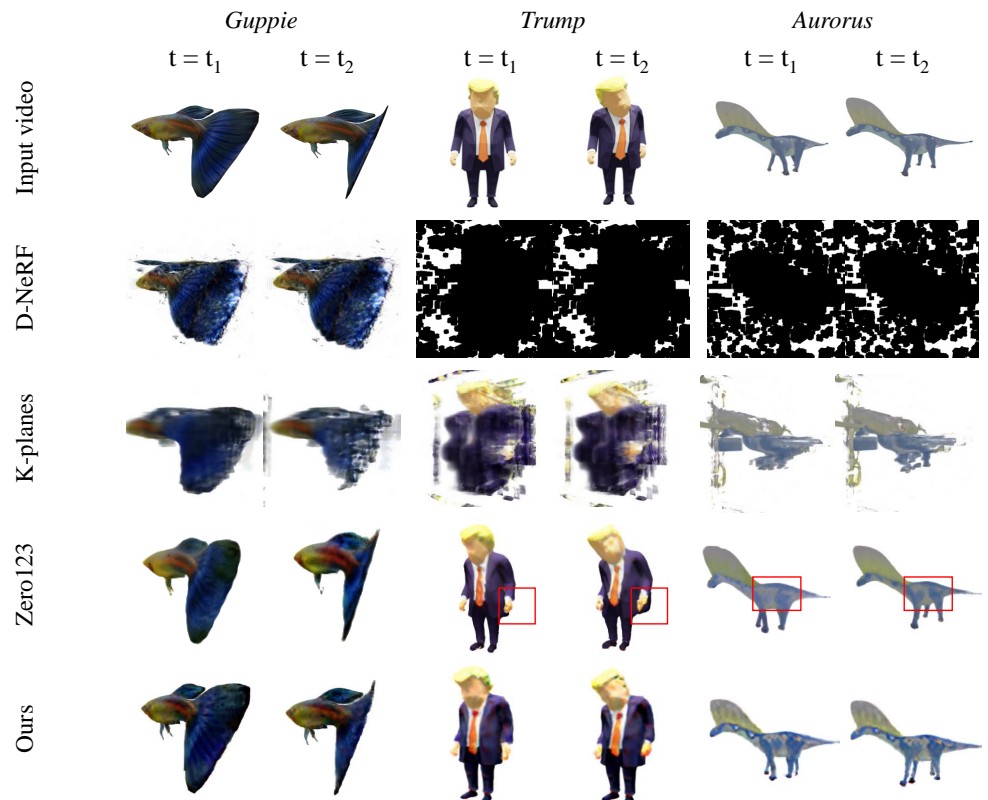

Figure 3: Comparison with dynamic NeRF methods and generative method. We render each dynamic object from a novel view at two timestamps. The temporal inconsistency in Zero123 is highlighted by the red box, but it's recommended to watch the video in supplementary material.

## 5.2 COMPARISONS WITH OTHER METHODS

To date, few methods have been developed for 4D generation utilizing video obtained from a static camera, so we compare our method with approaches with 4D modeling capabilities, i.e., D-NeRF (Pumarola et al., 2021) and K-planes (Fridovich-Keil et al., 2023), as well as approaches with 3d generation ability, i.e. Zero123 (Liu et al., 2023). For a fair comparison, video enhancer is not applied here.

**Quantitative Results** To quantitatively evaluate the proposed video-4D generation method, we provide image-level metrics, LPIPS (Zhang et al., 2018) and CLIP (Radford et al., 2021), as well as video-level metric, Frechet Video Distance (FVD) (Unterthiner et al., 2018). LPIPS and CLIP are computed between testing and rendered videos in a per-frame way, reflecting single-frame quality. FVD is computed between video pairs, taking both single-frame quality and temporal coherence in the entire video into consideration. We report the scores averaged over the four testing views of seven objects in Table. 1a (detailed metrics on each object can be found in the appendix). As shown in Table 1a, our dynamic 3D generation produces the best quantitative results over the other two dynamic NeRF methods on all metrics, which well aligns with the qualitative comparisons shown in Figure 3. Zero123 (per-frame reconstruction) has advantages over our method in terms of image-level metric, however, it lags behind the proposed method by a clear margin in terms of video-level metric FVD, which indicates severe temporal incoherence in Zero123 outputs.

**Qualitative Results** The outcomes of our method and those of other baselines are illustrated in Figure 3. Both D-NeRF and K-planes methods struggle to achieve satisfactory results in novel views, owing to the absence of multi-view information in the training data. Zero123, although outperforms ours in terms of image quality, suffers from severe temporal inconsistency, which could be observed via the video attached in the supplementary material. In contrast, our method manage to generate spatially and temporally coherent 4D object. Please refer to the appendix A.1 for additional results.

## 5.3 ABLATIONS

We perform ablation studies for every component within our framework. For clarity, the video enhancer is excluded when conducting all ablations except for its own. Quantitative results averaged on seven objects in the synthetic dataset are provided in Table 1b. Considering that the primary objective of introducing Cascade DyNeRF and ICL loss is to enhance spatial and temporal coherence, we advise readers to prioritize the video-level metric (FVD) over image-level metrics (LPIPS and CLIP) in evaluating this ablation study. The notable improvements observed in FVD scores underscore the efficacy of the proposed Cascade DyNeRF and ICL loss (as shown in the first four rows). The optional video enhancer improves the results slightly. Below, we present a qualitative analysis of Cascade DyNeRF and ICL loss on in-the-wild monocular videos, deferring the video enhancer results to the appendix due to limited space.

|  | w/o ICL | w/ ICL |
|---|---|---|
| preference rate(%) | 24.5 | **75.5** |

(a) Video-to-4D.

|  | w/o ICL | w/ ICL |
|---|---|---|
| success rate(%) | 19.3 | **28.6** |

(b) Text-to-3D.

Table 2: User study of Interpolation-driven Consistency Loss.

**Cascade DyNeRF** In Figure 10 (see in appendix), we conduct an ablation study for Cascade DyNeRF. Specifically, we substitute Cascade DyNeRF with the original K-planes architecture, maintaining all other settings unchanged. In the absence of the cascade architecture, the training proves to be unstable, occasionally yielding incomplete or blurry objects, as demonstrated by the first and second objects in Figure 10. In some cases, while the model manages to generate a complete object, the moving parts of the object lack clarity, exemplified by the leg and beak of the bird. Conversely, the proposed Cascade DyNeRF exhibits stable training, leading to relatively satisfactory generation results.

**Interpolation-driven Consistency Loss** The introduction of Interpolation-driven Consistency Loss (ICL) stands as a significant contribution of our work. Therefore, we conduct extensive experiments to investigate both its advantages and potential limitations. Figure 4a illustrates the ablation of both spatial and temporal Interpolation-driven Consistency Loss (ICL) in the video-to-4D task. Without ICL, the objects generated exhibit spatial and temporal inconsistency, as evidenced by the multi-face/foot issue in the blue jay and T-rex, and the unnatural pose of the seagull. Additionally, color discrepancies, such as the black backside of the corgi, are also noticeable. Employing either spatial or temporal ICL mitigates the multi-face issue, and notably, the use of spatial ICL also alleviates the color defect problem. Utilizing both spatial and temporal ICL concurrently yields superior results.

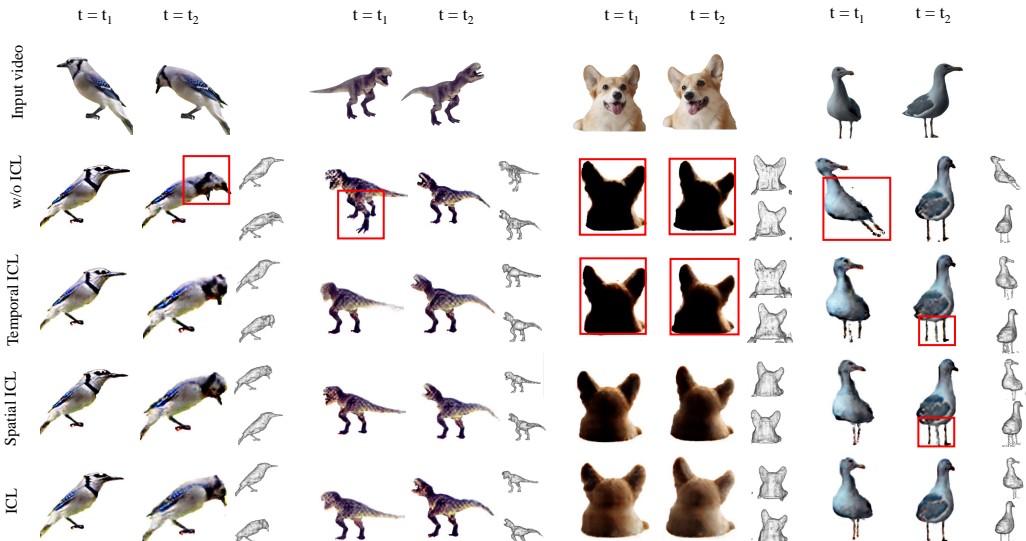

(a) Video-to-4D. For each dynamic object, we render it from a novel view for two timestamps with textureless rendering for each timestamp. For clarity, we describe the input videos as follows (from left to right): *blue jay pecking*, *T-rex roaring*, *corgi smiling*, *seagull turning around*.

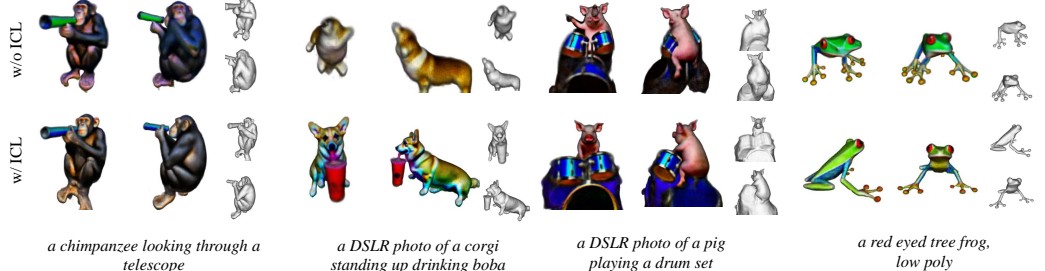

(b) Text-to-3D. For each 3D model, we render it from two views with a textureless rendering for each view and remove the background to focus on the actual 3D shape.

Figure 4: Ablation of Interpolation-driven Consistency Loss.

In the user study shown in Figure 2a, which averages the choices of 20 participants for 20 generated objects w/ and w/o ICL, 75% prefer the results w/ ICL.

We also test ICL loss on text-to-3D tasks. We collect all prompts related to animals from the official DreamFusion project page, totaling 230, and compare the success rate of DreamFusion implemented w/ and w/o the proposed ICL loss. The success rate comparison in Table 2b indicates results w/ ICL always outperform results w/o it. Qualitative comparisons are presented in Figure 4b which indicates the proposed technique effectively alleviates the multi-face Janus problem and thus promotes the success rate. Details could be found in the appendix.

## 6 CONCLUSION

We introduce a novel video-to-4d framework, named Consistent4D, aimed at generating 360° 4D objects from a single-view uncalibrated video. We first develop a Cascade DyNeRF to facilitate stable training under the discrete supervisory signals provided by an image-to-image diffusion model. More crucially, we introduce an Interpolation-driven Consistency Loss to enhance spatial and temporal consistency in 4D generation tasks. At last, a lightweight video enhancer is provided as an optional post-processing step. Comprehensive experiments conducted on both synthetic and in-the-wild data demonstrate the effectiveness of our method.

**Acknowledgements.** The authors would like to thank the anonymous reviewers for their valuable comments and suggestions. This work was supported in part by the Beijing Natural Science Foundation (Grant No. JQ22014, L223003), the Natural Science Foundation of China (Grant No. U22B2056, 62192782, 62036011, U2033210, 62102417), the Project of Beijing Science and technology Committee (Project No. Z231100005923046). Jin Gao was also supported in part by the Youth Innovation Promotion Association, CAS.

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

# A APPENDIX

## A.1 ADDITIONAL VISUALIZATION RESULTS

In Figure 5, we present the result of our method on four in-the-wild videos. For clarity, we describe the input videos as follows: *robot dancing*, *squirrel feeding*, *toy-spiderman dancing*, *toy-rabbit deforming*. Due to limited space, the reviewers are strongly recommended to watch the video in the attached files to see various visualization results.

## A.2 ABLATION ON CROSS-FRAME VIDEO ENHANCER

In Figure 11, we show the proposed cross-frame video enhancer could improve uneven color distribution and smooth out the rough edges, as shown in almost all figures, and remove some floaters, as indicated by the cat in the red and green box.

## A.3 DATA USED IN VIDEO-TO-4D QUANTITATIVE EVALUATION

| | Pistol | | | Guppie | | | Croco. | | | Monst. | | | Skull | | | Trump | | | Aurorus | | | Average | | |
|---|---|---|---|---|---|---|---|---|---|---|---|---|---|---|---|---|---|---|---|---|---|---|---|---|
| | LPIPS↓ | CLIP↑ | FVD↓ | LPIPS↓ | CLIP↑ | FVD↓ | LPIPS↓ | CLIP↑ | FVD↓ | LPIPS↓ | CLIP↑ | FVD↓ | LPIPS↓ | CLIP↑ | FVD↓ | LPIPS↓ | CLIP↑ | FVD↓ | LPIPS↓ | CLIP↑ | FVD↓ | LPIPS↓ | CLIP↑ | FVD↓ |
| D-NeRF | 0.52 | 0.66 | 1342.82 | 0.32 | 0.76 | 2244.47 | 0.54 | 0.61 | 2628.77 | 0.52 | 0.79 | 2720.27 | 0.53 | 0.72 | 3344.38 | 0.55 | 0.60 | 2145.56 | 0.56 | 0.66 | 1868.54 | 0.51 | 0.68 | 2327.83 |
| K-planes | 0.40 | 0.74 | 2060.83 | 0.29 | 0.75 | 2077.25 | 0.19 | 0.75 | 1823.25 | 0.47 | 0.73 | 2738.59 | 0.41 | 0.72 | 3338.74 | 0.51 | 0.66 | 3338.74 | 0.37 | 0.67 | 1304.83 | 0.38 | 0.72 | 2295.68 |
| Zero123 | **0.10** | 0.92 | 647.08 | **0.12** | 0.88 | 931.23 | 0.11 | 0.85 | 2038.22 | 0.16 | 0.93 | 2288.40 | 0.15 | 0.95 | 2490.25 | **0.23** | 0.88 | 1630.89 | **0.17** | 0.87 | 975.11 | 0.15 | 0.90 | 1571.60 |
| ours (s=2) | **0.10** | 0.90 | 853.89 | **0.12** | 0.90 | **811.23** | 0.12 | 0.82 | **1237.29** | 0.18 | 0.90 | **1307.53** | 0.17 | 0.88 | **2000.20** | **0.23** | 0.85 | 704.13 | **0.17** | 0.85 | 1019.81 | 0.16 | 0.87 | **1133.44** |
| ours (s=4) | **0.08** | **0.91** | **618.23** | 0.11 | **0.92** | **697.89** | **0.10** | **0.88** | **1069.90** | **0.15** | **0.94** | **964.73** | **0.13** | **0.94** | 2086.76 | **0.16** | **0.91** | **670.00** | **0.12** | **0.88** | **840.74** | **0.12** | **0.91** | **992.61** |

Table 3: Details of video-to-4D quantitative comparison.

Sin three dynamic objects are shown in Figure 3, we only visualize the rest four here, as shown in Figure 6. The observation is similar to the results in the main paper. Additionally, we provide the details of quantitative comparison in Table 3.

## A.4 THE NUMBER OF FRAMES

For simplicity, we sample each input video to 32 frames in all experiments. However, we find input videos without sampling sometimes give slightly better results, as shown in Figure 7.

## A.5 IMPLEMENTATION DETAILS

### A.5.1 CASCADE DYNERF

**Initialization** We follow Maigc3D (Lin et al., 2023) to initialize the dynamic NeRF. Specifically, the blob scale and standard deviation of the density are set as 10.0 and 0.5. The activation function is softplus.

**Optimization** We optimize the Dynamic NeRF using Equation 8, where $\lambda_1 = 0.1$, $\lambda_2 = 2500$, $\lambda_3 = 500$, $\lambda_4 = 50$, $_5 = 2.0$, and $\lambda_6$ is initially 1 and increased to 20 linearly until 5000 iterations. When applying ICL loss, we sample consecutive temporal frames at intervals of one frame and sample consecutive spatial frames at angular intervals of $5°$-$15°$ in azimuth. Reconstruction loss and foreground mask, alternate with ICL loss and SDS loss to optimize the model. When calculating SDS loss, the guidance scale of the diffusion model is set as 5, and the maximum/minimum percent of noise added to the rendering images decreases linearly from 0.98/0.8 at the beginning of the training to 0.25/0.2 at the medium of the training, and then kept unchanged.

### A.5.2 VIDEO ENHANCER

For video enhancer architecture, we follow pix2pix (Isola et al., 2017) except for that we modify the unet256 architecture to a light-weighted version, with only three up/down layers and one cross-frame attention layer. Our codebase for video enhancer is [1]the official GitHub repository of pix2pix, and we adopt all the default settings in their code except for Unet architecture, the learning rate and the training epochs, with the last two already mentioned in the main paper. For Unet(our video enhancer), the feature dimensions for the unet layers are set as 64, 128, and 256. Besides, we inject

---

[1]https://github.com/junyanz/pytorch-CycleGAN-and-pix2pix

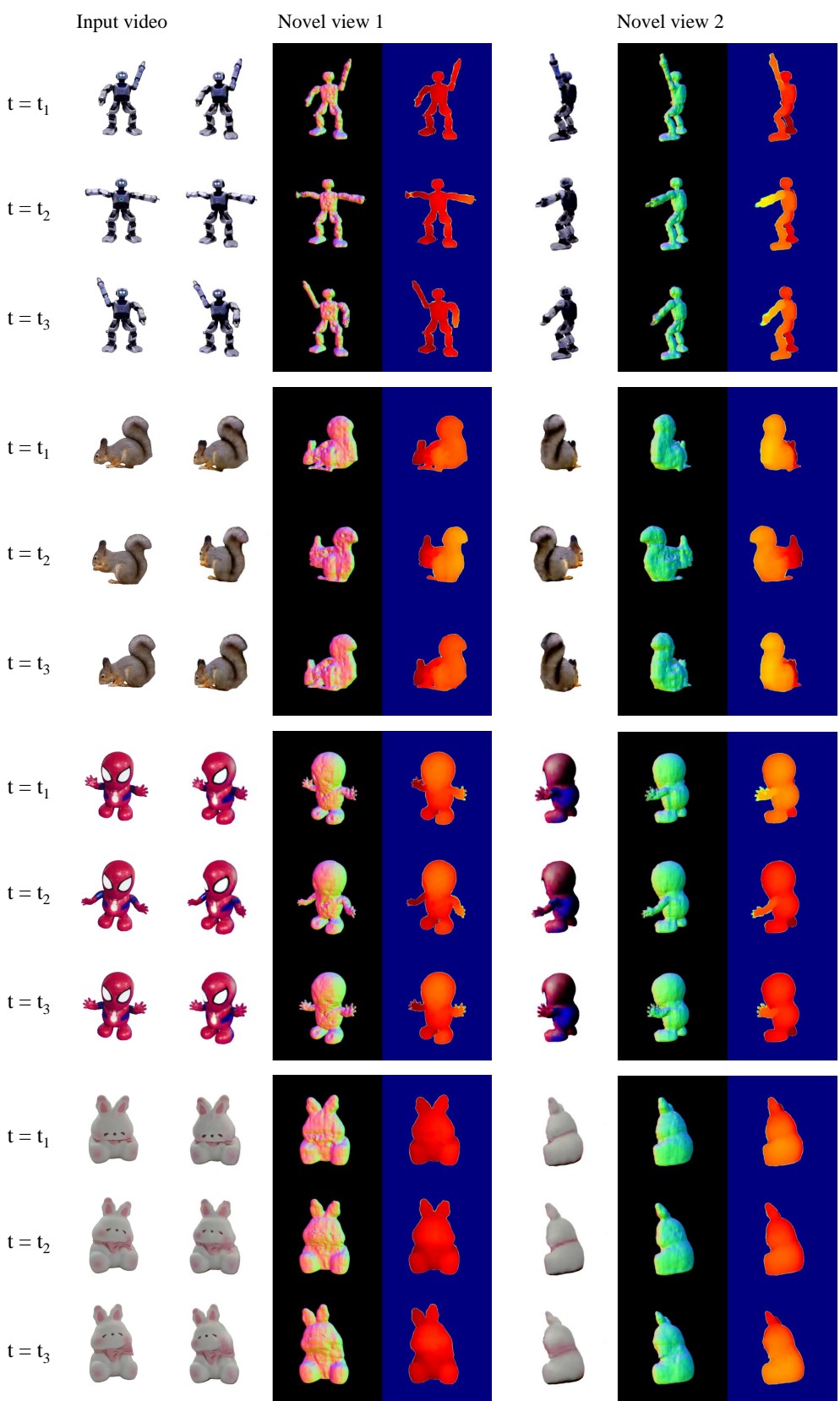

Figure 5: Visualization results of our method. All four input videos are in-the-wild videos. The novel views presented are 22.5° and 112.5° away from the input view, respectively. The results of our methods include RGB, normal map and depth map (from left to right).

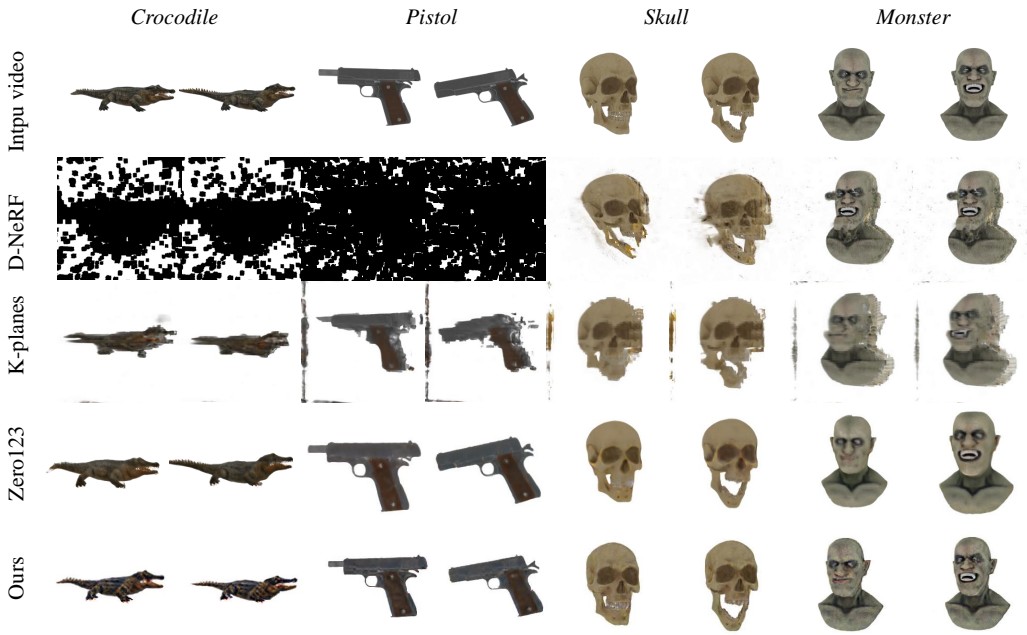

Figure 6: Data and comparison results for video-to-4D quantitative evaluation.

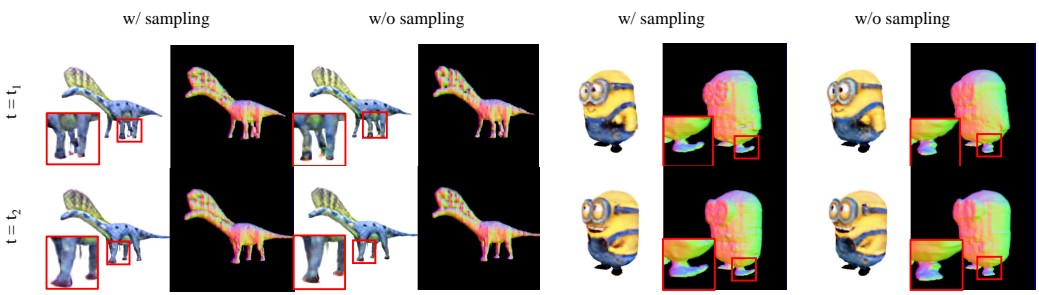

Figure 7: Ablation of video frame sampling. Videos w/ sampling contain 32 frames. Videos w/o sampling contain 72 and 39 frames for *Aurorus* and *minions*, respectively. The results of our methods include RGB and normal map (from left to right).

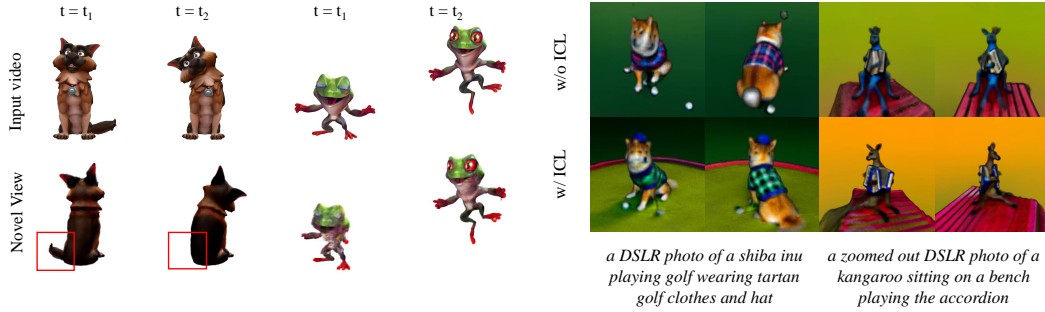

Figure 8: Video-to-4D failure cases.     Figure 9: Text-to-3D failure cases.

a cross-attention layer in the inner layer of the unet to enable the current frame to query information from adjacent frames. For generating real images, we use DeepFloyd-IF stage II (dee, 2023) in an image-to-image way (Meng et al., 2021) with denoising strength set as 0.35. Since this model is a diffusion model designed for single-image super-resolution, its outputs are images with improved quality yet in lack of temporal coherence. The input image, i.e., the rendered image, is resized to $64 \times 64$ and the output resolution is $256 \times 256$. The prompt needed by the diffusion model is manually set, i.e., we use the "a $*$" as the prompt, in which $*$ is the category of the dynamic object. For example, the prompts for dynamic objects in Figure 11 are *a bird*, *a cat*, *a minions*. The prompt cloud also be obtained from image or video caption models, or large language models.

### A.5.3   TEXT-TO-3D DETAILS

We choose Threestudio built by (Guo et al., 2023) as the codebase since it is the best public implementation we could find. DeepFloy-IF (dee, 2023) is employed as the diffusion model, and all default tricks in Threestudio are utilized. The hyper-parameters for results w/ and w/o ICL, such as batch size and learning rate, are kept consistent between the implementations w/ and w/o ICL, except for those related to ICL. We train the model for 5000 iterations, the first 1000 iterations with batch size 8 and resolution 64, and the rest 4000 with batch size 2 and resolution 256. The learning rate is 0.01 and the optimizer is Adam, the same as the default setting in Threestudio. The ICL loss is applied in the first 1000 iterations with probability 30% and weight 2000.

### A.6   FAILURE CASES

**Video-to-4D** Since the video-to-4D task in this paper is very challenging, our method actually has many failure cases. For example, we fail to generate the dynamic object when the motion is complex or abrupt, as shown in Figure 8. In Figure 8, the dog's tail disappears in the second image because the tail is occluded in the input image when $t = t_2$. The frog, which is jumping up and down fast, gets blurry when $t = t_1$.

**Text-to-3D** When applying ICL in text-to-3D, we find some multi-face cases that could not be alleviated, and we show them in Figure 9.

### A.7   LIMITATIONS

Although the proposed method achieves promising results for 360° dynamic object generation, our method has the following limitations: 1) Our method relies on a pre-trained diffusion model, and this limits the generalization ability of our method. Particularly, since the diffusion model adopted in this work is trained on synthetic dataset, our model might have worse performance when the input image/video is from the real-world. 2) The performance of our model relies on the quality of input video. We find when the input video is noisy, our model might not be able to generate the dynamic object in the video. 3) The training of our model costs more than 2 hours per object, and the long training time might present a challenge for practical deployment.

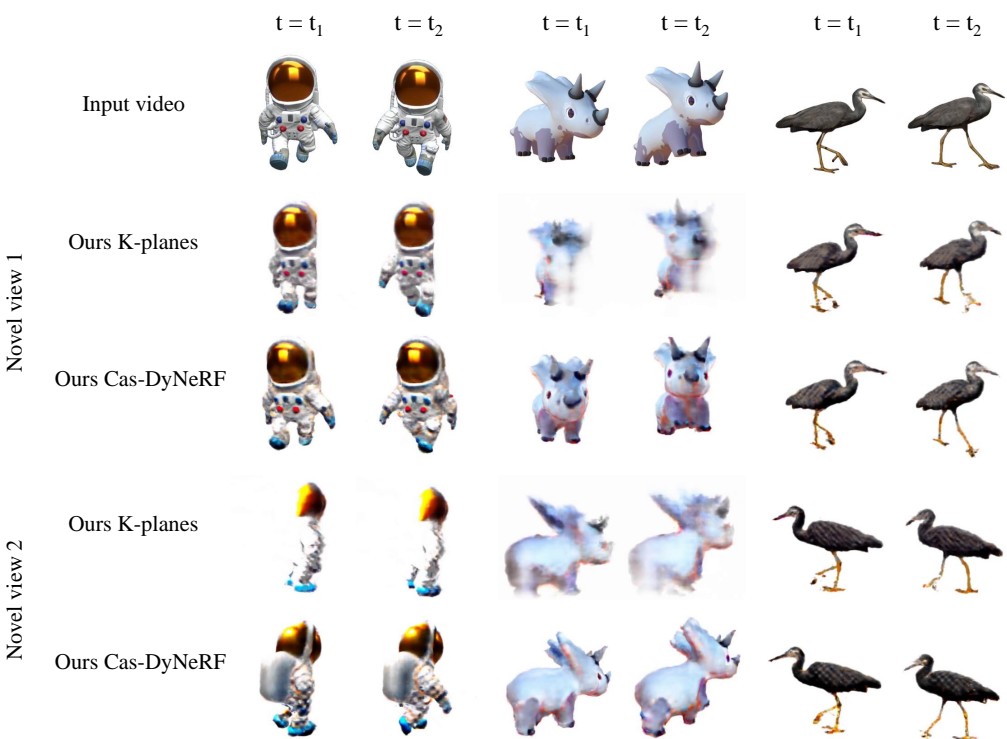

Figure 10: Ablation of Cascade DyNeRF.

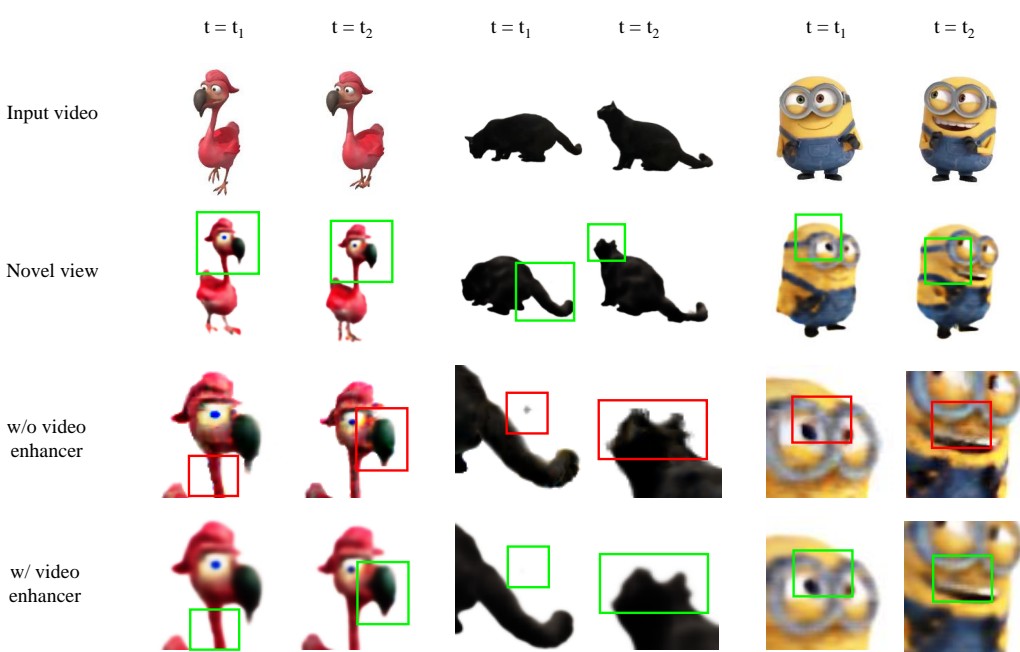

Figure 11: Ablation of Video enhancer. Please zoom in to view the details.

