# OpenReview forum: "Consistent4D: Consistent 360° Dynamic Object Generation from Monocular Video"
_ICLR.cc/2024/Conference — ICLR 2024 poster_

### Official Review · Reviewer_YMYg · 2023-10-31

**Soundness:** 4 excellent
**Presentation:** 4 excellent
**Contribution:** 3 good
**Rating:** 10
**Confidence:** 4

**Summary:**

The authors present a method by which a monocular, stationary video is turned into a 360 dynamic video of an object.  In particular, the authors first reconstruct a k-plane dynerf using SDS.  This is done in a hierarchical way so that low frequency spatio-temporal signal can be encoded first so that temporal continuity is preserved and then high frequency details are added afterwards.  An interpolation module is used to define losses between observed frames to further improve temporal continuity.  Finally, a video enhancement module uses a GAN to improve visual quality.

**Strengths:**

I think this is the first time that I've seen DyNerf combined with Dreamfusion.  The results look reasonable and the method doesn't appear overly convoluted.  The prior work section is sufficient to get the gist of what's going on and most of the bells and whistles have ablation studies.

**Weaknesses:**

I didn't really understand how the "real" images for the video enhancement GAN were generated.   There's some sort of super resolution network, but is that all the video enhancement network aims to do?  I didn't really understand why not just run this SR network on the generated images if you already have the network?  Is this a distillation step to accelerate the SR operation?  Or is the SR module only good for single image SR, but lacks temporal consistency?

The results in the supplemental have some pretty noticeable artifacts.  I think connecting DyNerf/k-planes + SDS is a pretty reasonable thing to do, but I'd hope for higher quality given the cascading approach.  In particular, the authors chose s=2 and don't increase this value.  It would be interesting to know how more layers would impact quality.

**Questions:**

Can the authors provide more justification for the video enhancer?  There are plenty of implementation details of the SR module in A.4, but I still would like to know what the main goal is in terms of training the GAN if you already have the SR module.

Has the cascaded dynerf been evaluated for s > 2?

---

> ### Author Response · Authors · 2023-11-21
> **Response to Reviewer YMYg**
>
> Dear Reviewer YMYg,
>
> We're grateful for your appreciation of the innovation and paper writing of our work, as well as your suggestions for improvement. Below is the response to your questions:
>
> * **The “real image” in video enhancement**:  The real image is generated by a pre-trained SR model. We use images output by the SR model as the “real image” in GAN because the SR model is designed for single images and **does not maintain temporal consistency across frames**. This lack of temporal consistency is challenging to detect in still images, so we attach a video of “real images” from the SR model as well as the output of our video enhancer in the supplementary material (video_enhancer.mp4). Please refer to it to see temporal inconsistency. We found training another lightweight Convnet as the video enhancer can alleviate this temporal inconsistency.
>
> * **Some noticeable artifacts in supplementary material**: We did not exclude all the failure cases when we provided the demo video in supplementary material, for example, the failure cases discussed in the failure case section (Sec. A.5) are also presented in the demo video. Some failures in our work stem from inaccurate supervision signals provided by the diffusion model. For instance, as seen in Fig. 9 of the appendix, the dog's tail occasionally vanishes. Other failures are linked to the flaws in our 4D representation and optimization processes, such as the blurring of the frog in Fig. 9.
>
> * **Increasing the cascade layers**:  Previously, we didn't increase the number of cascade layers because increasing it to 3 and 4 took around 45 and 60 GB of GPU memory for training, respectively, and we didn’t have access to computing resources with such large GPU memory. Now, we find Increasing cascade layers improves the texture of the generated dynamic object, alleviates the blurring effect, and facilitates spatial and temporal consistency. However, it can only slightly alleviate the negative effect of inaccurate supervision signal, which means failure cases caused by the diffusion model couldn’t be improved much by increasing cascade layers. Please refer to response 4 in “Responses to All Reviewers” for more details. Thank you for your suggestions.

---

### Official Review · Reviewer_B4LH · 2023-11-01

**Soundness:** 2 fair
**Presentation:** 3 good
**Contribution:** 3 good
**Rating:** 6
**Confidence:** 5

**Summary:**

This paper tackles the task of generating/reconstructing 4D objects from statically captured monocular videos: it utilizes L2 photometric loss and score-distillation sampling loss from pre-trained diffusion models. To push the quality, the authors exploit a cascaded mechanism and K-planes; to encourage consistency, the authors introduce interpolation-driven consistency loss. Qualitative and quantitative results demonstrate the effectiveness of the proposed approach.

**Strengths:**

The paper is easy to follow. The interpolation loss is interesting and seems effective.

**Weaknesses:**

### 1. Generative baselines

In the paper, the authors compare the proposed approach to D-NeRF and K-planes, which are regression-based methods. I think the authors should add at least one generative-based baseline to convince the effectiveness. Concretely, I think a valid baseline is to run a single-view reconstruction pipeline per frame [a] and see whether they can be consistent.

[a] https://github.com/threestudio-project/threestudio#zero-1-to-3-

### 2. About ablations

Currently, the only ablation in the paper is about whether to use interpolation consistency loss or not. Such ablation is only conducted with a user study.

I think there are quite a few ablations missing such that it is unclear which module contributes to the performance. To name a few (all could be provided with quantitative numbers):

a. How does each of the following loss contribute to the performance: foreground mask loss, normal orientation loss, and 3D smoothness loss

b. What if we remove the enhancer in Sec. 4.3.

c. Currently, the interpolation consistency loss is used with 25% probability during training. What if we have more or have less? It would be more informative if there could be a performance curve for this.

d. The weights for different losses differ quite a lot: the weight for reconstruction loss is 500 while the one for SDS is only 0.01. How the weight for SDS will change the performance?

### 3. About Baselines D-NeRF and K-planes

Frankly speaking, I am quite surprised about the inferior results of D-NeRF or K-planes in Fig. 3. Can the authors provide some discussion about this?

As mentioned in Bulletin 2: the proposed weight for SDS loss is only 0.01 while the weight for reconstruction loss is 500. Essentially, this is just a regression-based framework. Can authors clarify:

a. Do we apply the same reconstruction loss, including foreground mask loss, normal orientation loss, and 3D smoothness loss? If not, then the comparison seems misleading.

b. The difference between D-NeRF and K-planes: is the only difference is that D-NeRF uses MLP while K-planes use the plane representation?

### 4. Typo

Not sure whether this is a typo: should Eq. (6) be $\hat{x}_j = \phi (x_0, x_J, \gamma_j)$, i.e., the input is $x_J$ instead of $x_j$?

**Questions:**

See above.

---

> ### Author Response · Authors · 2023-11-21
> **Response to Reviewer B4LH (Part 1)**
>
> Dear Reviewer B4LH,
>
> Thank you for the recognition of the proposed ICL loss as well as the overall paper writing. Below is the detailed response to your concerns:
>
> First and foremost, we sincerely apologize for the typographical error in the SDS weight. **It should be 0.1 instead of 0.01**. We regret any confusion this mistake may have caused regarding our method.
>
> The hyperparameters of reconstruction weight and sds weight, i.e., 500 and 0.1,  are **exactly the same as the default config of Zero123 implemented in Threestudio**[1]. The two loss weights generate good results in Zero123, so we **did not finetune them in our experiment**. Other losses, including normal orientation loss and 3D smoothness loss, **were kept exactly unchanged in all experiments** in the paper to ensure a fair comparison.
>
> * **Generative Baseline**: We add zero123-per-frame as suggested, please refer to response 1 in “Responses to All Reviewers” for more details.
>
> * **Ablations**: **Previously, we presented extensive ablations for all of the novel modules introduced in our paper**, including Cascade DyNeRF, ICL, and the video enhancer, as detailed in **Sec. 5.3**. **Quantitative ablations** are newly added, as shown in response 2 in “Responses to All Reviewers”. For other commonly adopted losses in image-to-3D methods, such as the reconstruction loss and 3D normal smoothness loss, we maintained them as **default settings** and did not provide ablations. However, we are more than willing to conduct additional ablations to provide readers with a more comprehensive understanding of our work:
>    1. **The ablation of reconstruction loss, foreground mask loss, normal orientation loss and 3D smoothness loss**: While these losses have been adopted from Zero123 implemented in Threestudio and do not constitute our original contributions, we understand the importance of providing a comprehensive analysis. Below is the ablation for losses:
>
>       |  | LPIPS &darr; | CLIP &uarr; | FVD &darr; |
>       | -------- | -------- | -------- | -------- |
>       | ours w/o rgb loss    | 0.16   | 0.85   |  1260.39  |
>       | ours w/o mask loss    | **0.15**   | **0.87**   | 1157.97   |
>       | ours w/o normal orientation loss    | **0.15**   | 0.86   | 1146.39 |
>       | ours w/o 3d normal smooth loss    | 0.16  | 0.86| 1191.75 |
>       | ours w/o SDS loss    | 0.19  | 0.81 | 1699.83  |
>       | ours | 0.16 | **0.87** | **1133.44** |
>
>       It is evident from our metric analysis that the SDS loss holds a pivotal role, exerting a significantly greater impact compared to the RGB reconstruction loss. This observation underscores the generative nature of our method, emphasizing the importance of the SDS loss.
>
>
>    2. **Remove video enhancer**: The Video enhancer, while a valuable addition to our framework, does not constitute our core contribution. Its inclusion is aimed at enhancing the overall comprehensiveness of our approach. The improvement from the video enhancer is limited, as indicated in response 2 in “Responses to All Reviewers” (the last row of the table).
>
>    3. **Probability of ICL**: we provide the ablation of probability of ICL as bellow:
>       |  | LPIPS &darr; | CLIP &uarr; | FVD &darr; |
>       | -------- | -------- | -------- | --------
>       | probability=0.    |  **0.16** | **0.87**   | 1226.92 |
>       | probability=0.1    | **0.16** |  **0.87** | 1190.01   |
>       | probability=0.25 (ours) | **0.16**   | **0.87**   | **1133.44** |
>       | probabilityt=0.5    | **0.16** | 0.86 |  1179.67 |
>
>       FVD metric suggests that a probability value of 0.25 is an appropriate setting.
>
>    4. **SDS weight**: We provide experiments about changing sds weight as bellow:
>    |  | LPIPS &darr; | CLIP &uarr; | FVD &darr; |
>    | -------- | -------- | -------- | -------- |
>    | sds_weight=0.    |  0.19 |  0.81  |  1699.83   |
>    | sds_weight=0.01    | **0.16** | 0.85  | 1321.19  |
>    | sds_weight=0.1 (ours) | **0.16**   | **0.87**   | **1133.44** |
>    | sds_weight=1.    | **0.16** | 0.86 |  1270.86 |
>    The results indicate sds_weight=0.1 is a suitable setting.

---

> > ### Author Response · Authors · 2023-11-21
> > **Response to Reviewer B4LH (Part 2)**
> >
> > * **Bad performance of D-NeRF and K-planes**: D-NeRF and K-planes are reconstruction methods that rely on having sufficient multi-view information to reconstruct dynamic scenes effectively. However, the data used in this paper is a monocular video captured by a static camera, with **no multi-view information**. Consequently, D-NeRF and K-planes tend to **overfit the input view**, and when the viewing angle is slightly altered, they often fail to produce reasonable results. Furthermore, since their supervision signals are solely derived from the input view, these methods **struggle to reconstruct regions that are not visible in the training data**.
> >
> >    We are not surprised to see the failure of reconstruction by DyNeRF methods in our data setting, but we are surprised to see D-NeRF failed completely in some cases, i.e., the training loss just went to NaN. D-NeRF adopted a two-stage training, (as we mentioned in the following point), and the first stage training was normal (normal here means the overfitting to the input view). However, in the second stage, i.e., the point offset learning stage, **its training became extremely unstable**, despite we carefully finetuned the hyper-parameters. We conjecture that this instability arises from the inherent difficulty of predicting point trajectories in 3D, which necessitates supervision signals with multi-view information to mitigate uncertainty in 3D space.
> >
> > * **The difference between D-NeRF and K-planes**: Sorry that we did not introduce D-NeRF in the paper. Besides architectural differences, there is another noticeable difference between D-NeRF and K-planes. D-NeRF needs to first reconstruct a static object and then learn offsets of each point in future frames. In other words, it adopts a two-stage training. In contrast, K-planes doesn’t need to reconstruct the static scene first. It directly models the dynamic scene.
> >
> > * **Typo**: thanks for pointing it out, we have modified it.
> >
> > References
> >
> > [1] https://github.com/threestudio-project/threestudio/blob/main/configs/zero123.yaml#L129

---

> > > ### Comment · Reviewer_B4LH · 2023-11-22
> > > **Thank you**
> > >
> > > I appreciate authors's effort in addressing my concerns and questions. I am satisfied with the newly-added ablations, which make the paper more complete and convincing. I think they demonstrate the effectiveness of the proposed interpolation loss, which could be interesting and beneficial to the community. Hence, I increased my score.

---

### Official Review · Reviewer_Yv9V · 2023-11-03

**Soundness:** 3 good
**Presentation:** 3 good
**Contribution:** 3 good
**Rating:** 6
**Confidence:** 3

**Summary:**

The paper introduces "Consistent4D," a novel framework designed to generate 360° 4D dynamic objects from uncalibrated monocular video footage, a task with significant implications in the field of computer vision. This innovative approach is notable for its elimination of the traditional requirements for multi-view data collection and camera calibration, thus simplifying the 3D reconstruction process from dynamic scenes. The framework consists of two main components: a Dynamic Neural Radiance Field (DyNeRF) that creates a coherent structure from sparse input data, and a video enhancer that refines the output by addressing color inconsistencies and removing artifacts.

The authors validate their method through extensive experiments on various challenging datasets, including in-the-wild videos, demonstrating its robustness and versatility. The results indicate that Consistent4D can produce high-quality reconstructions with detailed textures and consistent geometry across different viewpoints and time frames. By showcasing successful reconstructions alongside cases where the method falls short, the paper provides a transparent view of the framework's current capabilities and potential areas for future improvement.

Overall, the contributions of this paper have the potential to significantly impact applications that rely on high-fidelity 3D dynamic object reconstruction, such as virtual reality, autonomous driving simulations, and digital content creation.

**Strengths:**

1、 The paper introduces a novel framework that significantly deviates from traditional multi-camera setups for 3D reconstruction, which is a unique approach in the field. The combination of Dynamic Neural Radiance Fields (DyNeRF) with a video enhancement process supervised by a 2D diffusion model and a GAN is an original contribution. This integration of techniques for addressing the 360° 4D dynamic object generation from monocular video footage is highly innovative.

2、The paper is structured in a manner that logically presents the problem, proposed solution, and validation, which enhances clarity and ease of understanding. The use of visual aids, such as figures and diagrams, is well-executed, providing a clear representation of the framework's functionality and the results obtained.

**Weaknesses:**

1、The figure 2 in this paper looks a little confusing, in terms of the input，output and loss propagation.

2、 More comparisons are needed in this setting, are there other datasets which can be evaluated on this tasks. Lack of training time and inference time comparison with other methods.

3、The detailed usage of the diffusion model in the method part are needed for better understanding this paper.

**Questions:**

see weaknesses

---

> ### Author Response · Authors · 2023-11-21
> **Response to Reviewer Yv9V**
>
> Dear Reviewer Yv9V,
>
> Thanks for the appreciation of the novelty as well as the paper writing of our work. Below is our detailed response to your questions:
>
> * **Explanation of Figure 2**:
>
>    a) **The input** is a monocular video captured by a static camera, and the image sequence from this video is used as the condition of diffusion models for SDS loss as well as the ground truth for reconstruction loss .
>
>    b) **The output** is a dynamic NeRF representing the dynamic object. Besides, we can use the video enhancer to further enhance the video rendered from dynamic NeRF.
>
>    c) **Loss propogation**: Our method contains two stages. In first stage, SDS loss provided by the diffusion model is used to supervise the training of dynamic NeRF through the rendering images. The proposed ICL loss is used to supervise the dynamic NeRF in the same way. In the second stage, we employ a GAN framework, with the generator acting as the video enhancer. We use the conventional GAN loss, and we apologize for an accidental error in the loss direction depicted in Figure 2. We have rectified this mistake.
>
> *  **Comparisons on other datasets**: Existing multi-view dynamic scene datasets are usually scene-level, however, currently our method can only tackle object-level generation. A few datasets are object-level, however, the dynamic objects in those datasets are so complicated that even single-image-to-3D methods fail to generate the static 3D object from a single frame in the video, let alone our methods for dynamic object generation.  We will improve our work and extend it to the scene level and complicated objects in the future.
>
> * **Training time and inference time comparison with other methods**:
> Since Zero123 requires more than 40GB GPU memory for training, we compare the training and inference time of all methods on **an 80G A100 GPU**. We provide the averaged results on 4 objects as follows (inference speed is measured under image resolution of **256x256**):
> |  | training time | inference speed |
> |-----------------|-----------------|-----------------|
> | D-NeRF   | 61 min   | 2.8 fps   |
> | K-planes   | 76 min   | 19.2 fps  |
> | Zero123(per-frame)   | 79 min   | 41.8 fps   |
> | ours w/o video enhancer   | 91 min  | 63.8 fps  |
> | ours w/ video enhancer   | 97 min  | 62.9 fps   |
>
> The training time of our method is a little longer than per-frame Zero123, whilst we have fast inference speed.
>
> * **Detailed usage of the diffusion model**: We illustrated the use of the diffusion model in the Preliminaries (Sec. 3.1)  and we elaborate on it below:
> The diffusion model adopted in this work is Zero123-xl, an image-to-image translation model. In the original design, this diffusion model takes a reference view image and relative camera position between the target view and the reference view as the input and outputs the target view image.  In our pipeline, we use images from the monocular video as the reference images. Following image-to-3D methods, we random sample target views for rendering images from dynamic NeRF, and calculate the relative position between the target views and the reference view to serve as the camera position input of the diffusion model. Then, the diffusion model can calculate SDS loss for the rendering images of the target view via Eq. 2 in the paper. The gradient is backward to dynamic NeRF through the rendering images.
>
>    For easy understanding, we now provide another interpretation of SDS loss in Eq. 2. SDS loss is equal to the L2 loss between the noisy image (obtained by adding random noise to rendered target view images) and the 1-step denoised image output by the diffusion model. Thus, the distribution of rendered images from target views is always optimized toward the distribution of target images in the diffusion model.

---

### Official Review · Reviewer_ZZjx · 2023-11-04

**Soundness:** 3 good
**Presentation:** 2 fair
**Contribution:** 3 good
**Rating:** 6
**Confidence:** 4

**Summary:**

The paper proposes Consistent4D, a novel framework for generating 4D dynamic objects (i.e. 360-degree views over time) from a single monocular video captured by a static camera. The work presents a novel approach for generating consistent 4D objects from monocular videos, demonstrating the promise of leveraging generative models for this challenging problem. The consistency loss helps improve both video-to-4D and text-to-3D generation.

**Strengths:**

The paper introduces an innovative approach to 4D object generation from monocular videos, focusing on generation instead of reconstruction, and uniquely utilizes generative models as priors. A novel consistency loss driven by video interpolation enhances spatiotemporal coherence. The method shows superior performance over baseline DyNeRF methods through extensive evaluation on synthetic and real datasets, demonstrating high-quality 4D outputs. The paper is well-structured and clearly articulated, with detailed method descriptions and thorough component ablations, effectively using tables and figures to convey results. This research represents a significant advancement in 4D content generation from monocular videos, with the potential of the introduced consistency loss to improve other generative 3D tasks. The paper is recognized as a strong contribution to the field, particularly for its consistency loss technique that could influence further generative 3D research.

**Weaknesses:**

The paper's potential weaknesses include a heavy reliance on pre-trained models that could limit flexibility in different scenarios, lack of detailed validation for generalizability and robustness which is crucial for assessing the method's practical applicability, and insufficient discussion on computational efficiency and scalability that are vital for real-time applications. Moreover, the absence of comprehensive quantitative and qualitative assessment metrics may impede a full understanding of the method's performance, especially when direct comparisons with state-of-the-art techniques and benchmarking are not thoroughly presented. Lastly, a detailed discussion of the technical limitations and potential failure cases would be beneficial for readers to understand the applicability and boundaries of the proposed method.

**Questions:**

What is the computational cost of your approach, and is it suitable for real-time applications? Can you provide benchmarks on the computational resources required?
Have any user studies been conducted to assess the qualitative aspects of the generated 4D objects? If not, what was the rationale behind this choice?
What are the main limitations of the proposed method, and in what scenarios does it tend to fail? How do these limitations affect the practical deployment of the method?

---

> ### Author Response · Authors · 2023-11-21
> **Response to Reviewer ZZjx**
>
> Dear Reviewer ZZjx,
>
> Thank you for acknowledging the significant contribution of our work to the field, especially the proposed ICL loss. Below, we try to carefully address each of your concerns:
>
> * **Reliance on pre-trained models**: our method necessitates the use of pre-trained diffusion models to generate supervision signals for 4D generation. This is crucial for ensuring the generalization ability of the 4D generation task as only limited 4D data is publicly available for training.
>
> * **Computational efficiency**: The training time of dynamic NeRF and video enhancer is 2.5 hours and 15 minutes respectively on a single V100 GPU. Please refer to response 3 in “Responses to All Reviewers” for more details. Our method is not capable of real-time application at this juncture, however, we are optimistic about achieving real-time efficiency in the future.
>
> * **Comprehensive comparison**:
>    a) dataset: Previous multi-view dynamic scene datasets are mainly scene-level, but our method can only tackle object-level generation. So we render multi-view video from animated modes by ourselves for quantitative evaluation. We will improve and extend our work to scene level in the future.
>
>    b) comparison methods: Since there is **no previous method capable of both zero-shot 3D generation and temporal modeling**, our comparison includes dyNeRFs, which excel in temporal modeling but lack generation capability, and Zero123, which offers generation ability but lacks temporal modeling. Please refer to **responses 1 and 2 in “Responses to All Reviewers”** for quantitative results.
>
> * **Limitations and potential failure cases**: Thanks for pointing this out. We previously provided failure cases in the appendix section (Sec. A.5), and we have elaborated the limitations in the updated appendix as suggested (Sec. A.6) . A brief summary of limitations of the current approach is as follows:
>    1. Easy to fail when the object movement is complicated and fast
>    2. Easy to fail when the input video is blurred or noisy
>    3. Easy to fail when the foreground segmentation is not good
>    4. Easy to fail when the input video contains multiple foreground objects
>    5. Rely on the pre-trained diffusion model.
>    6. Cannot be real-time
>    7. Cannot handle scene-level reconstruction
>
> * **The effect of limitations on practical deployment**:
>    1. The efficacy of our model is closely tied to the quality of the input video. In practical deployments, it may struggle to accurately reconstruct dynamic objects when faced with low-quality input videos
>    2. The dependency on a pre-trained diffusion model can adversely affect the generalization capability of our approach. This concern is particularly pronounced given that the diffusion model employed in our work is trained on synthetic data. Consequently, there's a potential for a domain gap when processing real-world input images or videos. Our experiments indicate a higher likelihood of reconstruction failures with real-world videos as compared to those sourced from animated films
>    3. The relatively extended training time of our model presents a challenge for its practical deployment
>
>    We will try to improve our work to address the above limitations in the future.
>
> * **User study**
>    1. We are **the first work** aimed at 4D dynamic object generation from uncalibrated monocular video, so it’s difficult to find proper methods to compare with. Current comparison methods include two dynamic NeRF methods (previously provided by us) and one image-to-3D method (suggested by the reviewers, run in a per-frame reconstruction way). The former suffers from failure reconstructions and the latter results in severe temporal inconsistency, as shown in qualitative and quantitative results in Figure 3, Table 1, and the visualization video in the supplementary material (comparison_zero123_baseline.mp4).  **Obvious bad performances are observed in all comparison methods given that they are not specially designed for this task.** Thus, we regard a user study between them and our method as unnecessary.
>    2. Instead, we previously provided **two user studies for the ICL loss**, the key contribution of our work. Please refer to Sec.5.3, Table 2 in the main paper for details.

---

### Official Review · Reviewer_6R15 · 2023-11-04

**Soundness:** 2 fair
**Presentation:** 2 fair
**Contribution:** 2 fair
**Rating:** 6
**Confidence:** 4

**Summary:**

This paper models Monocular Dynamic Reconstruction problem as a consistent 4D generation. Given an input image sequence the authors optimise for Dynamic Radiance field using score distillation sampling form a pre-trained diffusion model while enforcing spatial and temporal view synthesis consistencies via an video interpolator. A cascade version of K-palnes is used for modelling dynamic radiance fields and a video enhancer module is utilised to improve quality.

**Strengths:**

The paper assembles latest advances in dynamic NeRFs and image guided 3D generation to create novel pipeline for consistent 4D reconstruction where state of art frame interpolation techniques play key role in enforcing consistency on single view 3D generation. The contribution has been evaluated via ablation study independently. The pipeline is sufficiently innovative for and results are impressive enough to be published in ICLR.

Additionally, few architectural innovations in terms of modelling Dynamic NeRFs can benefit the field focusing on view synthesis via geometric cues as well.

**Weaknesses:**

1. Writing of the paper can improve a lot. Paper's readability is significantly hampered due to informal writing style, usage of obscure terminology and grammatical errors. Crucial details are left out leaving reader to fill gaps by extrapolating prior art. Also, notations are misplaced in a few places. (I did not check all but see questions section for the details on these)
2. Some of the experimental evaluation, especially quantitative evaluation is not appropriate and given synthetic datasets are available on which authors evaluate the proposed method already, the same can be used for more appropriate Comparisions and ablations.
3.  Due to lack of details, over complicated setup involving too many neural networks and very unclear writing, the experiments presented in the paper are not reproducible in my opinion. See details and suggestions in the questions section.

**Questions:**

1. Suggestion on paper's structure: The paper attempt to focus on too many incremental changes within individual components the 4D reconstruction pipeline uses. Some of these changes make no significant difference in the final results e.g. video enhancer though. Trying to pack and justify these incremental changes is difficult in limited space to which author did not do justice. I would highly recommend leaving some of these details out to appendix and incorporate more critical explanation of the pipeline -- which I reiterate is sufficiently novel for publication once explained clearly. Things such as loss function used to train the entire system must be included in the main body of the paper and clearly written. I would also highly recommend avoiding a story telling narrative currently used in the paper where a
 series of failed/suboptimal experiments are listed one after the other without letting the reader know what is eventually used. Instead, explaining what the proposed pipeline and methodology is, followed by reiterating novel changes will be more clear.  Incremental changes in K-plane with feature concatenations, cascade DyNerf justification followed by by statement that Cascade DyNeRF alone is not enough for spatiotemporal consistency, so we resort to extra regularization, please see in the next section" confuses reader regarding what is novel.

2. The paper uses multiple off the shelf deep networks within an iterative optimisation framework to train for radiance file, a video enhancer and GAN. Technical details for most of which is omitted even from the appendix making reproducing the experiment impossible. Authors should also comment on reconstruction times and compute required to generate the results.

3. While the reconstructions looks impressive, I am not convinced with the quantitative evaluations presented in the paper. First, It is not fair to compare the proposed approach with Dynamic NeRF literature given that the baselines are designed to work on videos with sufficient camera motion only and not use many foundation models. As the proposed method introduces the 4D consistency via video interpolation module in 3D generation frameworks, would it not be more suitable to quantitatively evaluate the results with a baseline like Zero123? Ablations of ICL gives a flavour of the same but it is only qualitative. I would strongly recommend the 4D consistency and accuracy of the method is quantitatively evaluated on the synthetic sequence.

4. Besides, given that the authors use the image  reconstruction loss used in NeRF (I struggled to verify the same), I disagree that traditional novel view synthesis losses (PSNR,SSIM) should be avoided. In fact, I do wonder why some the generated novel views for example T-rex roar have Seagull turning around have texture don't have texture consistent with the input views. Perhaps the authors can elaborate. In fact a through 4D reconstructions evaluation requires not only good novel view synthesis (consistent with input appearance) but a good quality key-point tracking as well as accurate structure. Given that the authors have used synthetic datasets some of this can be evaluated as well. I understand that the paper focuses on the challenging scenario of stationary camera for 4D reconstructions but in principle the methodology can be evaluated on any sequence Dynamic NeRFS are evaluated on and encourage authors to do so.

Some of informal writing and notational inconsistency requiring explaination can be find below. These are non exclusive:

a. “DyNeRF methods mainly assume the supervision signals are temporally coherent, however, this assumption does not hold in our task,” The author never formally defines what is "temporal coherence" However the task paper solve is same as that of Dynamic NeRF. i.e. reconstructing deforming scene.  Perhaps authors are tring to allude to the challenges in incorporating diffusion models as priors which have no notion of time?

b. “In order to minimize the impact of temporal discontinuity in the supervision signals, we are prone to DyNeRF methods with low-rankness”, Are authors suggesting they prefer to use a framework like K-planes – which imposes rank constraints - as the suitable dynamic NeRF framework? It is unclear to me why low rank will specifically help the proposed approach beyond being a useful prior for 4D reconstruction is general case. Please specify.

c. Equation 6: Does the indices j and batch size J are being confused in the equation? I suppose that a sensible interpolation loss will take image x_1 and x_J as input with \gamma_j and return the image \hat{x_j}? Also, while the text describe the input images to be indexed 1 and J the equation itself use x_0 and x_j? The author does not differentiate between spatial and temporal constancy. Is the spatial consistency enforces 3D consistency of rendered views using virtual cameras at a single time or something else?

d. “The spatial and temporal resolution of Cascade DyNeRF are configured to 50 and 8 for coarse-level, and 100 and 16 for fine-level, respectively. We first train DyNeRF with batch size 4 and resolution 64 for 5000 iterations. Then we decrease the batch size to 1 and increase the resolution to 256 for the next 5000 iteration training” I am not certain what the resolution means in the second statement. Technical details in this paragraph are very hard to understand in general and can be explained better.

e. “we also apply foreground mask loss, normal orientation loss, and 3D smoothness loss.” These losses should be explained in appendix (i prefer within the body of the paper) and cited in the paper if they are coming from literature directly.

---

> ### Author Response · Authors · 2023-11-21
> **Response to Reviewer6R15**
>
> Dear Reviewer 6R15,
>
> Thanks for the acknowledgment of the innovation and effectiveness of our framework. Below, we provide detailed responses to address your concerns:
>
> * **Paper writing**: we have re-origanize the paper as you suggested. We list the changes we made below:
>
>   1) Confusing sentences about Cascade DyNeRF are removed
>   2) Video enhancement is claimed as an optional step
>   3) Loss function part is moved to the main body of the paper
>   4) **Comprehensive implementation details** regarding both DyNeRF training and video enhancer training are provided in the appendix (Sec. A.4)
>   5) Zero123 is added in comparison with state-of-the-art, and quantitative ablations are added as well
>
> * **Training time**: Training time for dynamic NeRF/video enhancer is around 2.5 hours/15 minutes, measured on a single V100 GPU. For details, please refer to response 3 in “Responses to All Reviewers”.
>
> * **Comparison with Zero123**: Zero123 lags behind us in terms of temporal consistency. Please refer to response 1 in “Responses to All Reviewers” for more details.
>
> * **Quantitative ablation**: The improvements in the metric FVD validate the effectiveness of each proposed module. Please refer to response 2 in “Responses to All Reviewers” for more details.
>
> * **Reproducibility**: We have provided extensive implementation details, please refer to response 5 in “Responses to All Reviewers”.
>
> * **PSNR and SSIM as losses**: Thanks for suggesting using the two losses. Actually, we had conducted experiments using L1/PSNR/SSIM as the rendering loss. We empirically find that the L2 loss generates the best result,  and that is the reason for choosing the L2 loss in our method.
>
> * **Inconsistent texture**: The texture inconsistency problem is widely observed in generative image-to-3D studies [1,2,3,4], which might be related to the SDS optimization. It will be our future work to improve the texture consistency.
>
> * **Key-point tracking**: Currently, our model employs an implicit 4D field to model dynamic objects, which precludes the possibility of key-point tracking. This could be potentially addressed by using a point-based 4D model (e.g., dynamic 3D Gaussians) and we leave it as a  future work.
>
> * **Evaluation on previous dynamic NeRF settings**: Although conventional reconstruction is not the focus of the proposed method, we are glad to test our model in that setting. We will do this experiment.
>
> * **Explanation**
>
>    a）Temporal coherence as the main challenge:  You’re right. Temporal coherence is a unique challenge in our pipeline, cause we adopt an image diffusion model without the notion of time to provide the supervision signal.
>
>    b)  K-planes with low-rankness: Thanks for pointing out. We agree that the use of “low-rankness” is inaccurate and confusing, so we have removed it in the revised paper. What we want to express is, K-planes is naturally prone to temporal coherence given that it has temporal interpolation operation. Considering temporal coherence is the main challenge in our pipeline, we choose to build our dyNeRF on K-planes.
>
>    c) Typos in Eq.6 and questions about spatial consistency: Thanks for pointing out. Yes, you’re right about Eq.6 and we have fixed those typos. For spatial consistency, we render batch images using virtual cameras at a single timestamp when applying ICL loss.
>
>    d) The meaning of resolution in implementation details: The resolution in the second statement stands for the resolution of rendering images during training. Due to limited GPU memory, we can only render 64x64 images when using batch_size=4.
>
>    e)  Loss functions: We have moved them to the main body of the paper and have cited corresponding literatures in the revised paper.
>
>
> References
>
>
> [1] Nerdi: Single-view nerf synthesis with language-guided diffusion as general image priors. In CVPR, 2023.
>
> [2] Realfusion: 360deg reconstruction of any object from a single image. In CVPR, 2023.
>
> [3] Make-it-3d: High-fidelity 3d creation from a single image with diffusion prior. In ICCV, 2023.
>
> [4] Dreamgaussian: Generative gaussian splatting for efficient 3d content creation. In arXiv, 2023.

---

### Author Response · Authors · 2023-11-21
**Responses to All Reviews (Part 1)**

We thank all reviewers for their insightful comments. We are encouraged to hear that our work “is (highly) innovative” (R1, R2, R3, R5), “represents a significant advance in 4D content generation from monocular video” (R2), “results are impressive” (R1), and “the proposed ICL loss is effective” (R1, R2, R4). Meanwhile, we feel sorry for the confusion caused by typos, unclear expressions, and potentially inadequate comparisons.

Regarding comparison with other methods, to the best of our knowledge, our work is **the first and the only work** targeting at the task of 360-degree dynamic object generation from uncalibrated monocular video. Given this, we did not find a proper baseline when submitting the paper, and we appreciate the reviewers for suggesting the baseline of per-frame Zero123. Below we try to address the common concerns raised by reviewers:

1. **Zero123 (per-frame) as a comparison baseline**

Previously, we selected D-NeRF and K-planes to serve as baselines due to their capabilities in temporal modeling. As suggested by reviewers, we additionally compare our method with the separate per-frame Zero123 generation. To make the results more convincing, we apply the Zero123 implementation in Threestudio[1] and adopt the corresponding default config. In terms of evaluation metrics, the commonly used Frechet Video Distance (FVD) [2] in video generation tasks [3,4,5,6] is applied, which measures both the per-frame image quality and the temporal coherence of the whole video. Quantitative comparisons between D-NeRF, K-Planes, per-frame Zero123, and our Consistent4D, are listed below (the lower FVD, the better performance):

|Method\Object  | Pistol | Guppie | Crocodile | Monster | Skull | Trump | Aurorus | Average |
|----------|----------|----------|----------|----------|----------|----------|----------|----------|
| D-NeRF   | 1342.82    | 2244.47  |  2628.77   | 2720.27   | 3344.38 |  2145.56   | 1868.54   |  2327.83   |
| K-planes  | 2060.83   | 2077.25  |  1823.25   | 2738.59    |  3338.74  | 2726.28 | 1304.83 |  2295.68    |
| Zero123(per-frame)   | **647.08**    | 931.23  | 2038.22   | 2288.40    |  2490.25 |  1630.89   | **975.11**   |  1571.60    |
| Ours    | 853.89    | **811.23**    | **1237.29**   | **1307.53**    | **2000.20**    | **704.13**    | 1019.81    |**1133.44**    |

We find that Zero123 significantly underperforms compared to our method in the video-level metric of FVD. This discrepancy suggests **severe temporal incoherence** in the per-frame Zero123 setting. For more comprehensive comparisons, we refer reviewers to our revised supplementary material on video visualizations of the per-frame Zero123 generation (comparison_zero123_baseline.mp4).

2. **Quantitative ablations**

 Following the suggestion, we have performed quantitative ablation studies on the synthetic dataset utilized in our comparison with state-of-the-art methods. We report the average metrics of the image-level LPIPS/CLIP and the video-level FVD across seven objects as follows:

| Cascade DyNeRF | ICL | Video Enhancer | LPIPS &darr; | CLIP &uarr; | FVD &darr; |
|:----------:|:----------:|:----------:|----------|----------|----------|
|   x  |   x  |   x  | 0.16    | 0.86    | 1303.31    |
| √    | x    | x    | 0.16    | 0.87    | 1226.92    |
| x    | √     | x    | **0.15**    | **0.88**    | 1205.80    |
| √    | √    | x    | 0.16    | 0.87    | **1133.44**   |
| √     | √     | √     | 0.16    | 0.87    |   **1114.85**  |

While different settings produce similar LPIPS and CLIP scores, it is noteworthy that each component **subsequently improves the FVD score**. The quantitative results well align with our primary objective of introducing cascade DyNeRF/ICL/video enhancer to enhance the spatial and temporal coherence in 4D generation.

3. **Training and computing resources**

The initial training time per object reported in our paper was around 6 hours for the dyNeRF part and 15 minutes for the video enhancer part, on a single V100 GPU (See implementation details in Sec. 5.1). Recently, we optimized the data loading pipeline by stashing the video to avoid repeated image loading in every training iteration, which reduces the training time of dyNeRF from 6 hours to 2.5 hours. The training time of the video enhancer remains the same.

---

### Author Response · Authors · 2023-11-21
**Responses to All Reviews (Part 2)**

4. **Better performance when increasing Cascade Dynamic NeRF layers**

As suggested by Reviewer YMYg, we do experiments about increasing the number of cascade Dynamic NeRF layers and we find increasing layers leads to better results. Averaged results on synthetic datasets are reported as below:

|  | LPIPS &darr; | CLIP &uarr; | FVD &darr; |
| -------- | -------- | -------- | -------- |
| layer=2    | 0.16   | 0.87   | 1133.44 |
| layer=3    | 0.15 | 0.87  | 1084.60   |
| layer=4    |  **0.14**         |  **0.90**        |  **992.61**          |

We find that the increased Cas-DyNeRF layers can lead to better results in terms of both image-level quality and video-level temporal consistency. We also provide qualitative comparisons **in the supplementary material** (**cascade_layer.mp4**)

5. **Reproducibility**

We have updated all implementation details in the appendix Sec. A.4. Moreover, we have provided extensive implementation details there, including pseudo-code for Cascade DyNeRF, figures for video enhancer architecture, etc. The code and data used in this work will soon be made publicly available for reproducing the results and improvement over the paper.


We have updated the paper and the appendix according to the suggestions of the reviewers. Particularly, we include **a diff mode version of the paper in the supplementary material** (Consistent4D_ICLR_rebuttal_diff.pdf) for a clear understanding of the changes we have made. (Deletions and additions are indicated by blue and red, respectively)


References


[1] threestudio: A unified framework for 3D content generation. (https://github.com/threestudio-project/threestudio）

[2] Towards accurate generative models of video: A new metric & challenges. In arXiv, 2018.

[3] Stylegan-v: A continuous video generator with the price, image quality and perks of stylegan2. In CVPR,  2022.

[4] Generating Long Videos of Dynamic Scenes. In NeurIPS, 2022.

[5] Phenaki: Variable Length Video Generation from Open Domain Textual Descriptions. In ICLR, 2023.

[6] CogVideo: Large-scale Pretraining for Text-to-Video Generation via Transformers. In ICLR, 2023.

[7] Quo vadis, action recognition? a new model and the kinetics dataset. In CVPR, 2017.

---

### Meta-Review · Area_Chair_31nB · 2023-12-13

**Metareview:**

The paper presents a novel framework for generating 4D dynamic objects (i.e., 360-degree views over time) from a single monocular video captured by a static camera. This work proposes modeling the monocular dynamic reconstruction problem as a coherent 4D generation. It uses fractional distillation sampling with a pre-trained diffusion model to optimize the dynamic radiance field, while enforcing spatial and temporal view synthesis consistency through a video interpolator. This research represents a significant advancement in generating 4D content from monocular videos, with the introduced consistency loss potentially improving other generative 3D tasks. However, the proposed method lacks slightly in universality and robustness.

**Justification For Why Not Higher Score:**

Nevertheless, there are some shortcomings. Firstly, the writing is not standard enough, lacking readability and poor organization. Secondly, the model has poor generalizability and the evaluation is not sufficiently thorough.

**Justification For Why Not Lower Score:**

Experiments demonstrate the effectiveness of this paper's construct. All reviewers gave positive or slightly positive evaluations, therefore it is recommended to accept this paper.

---

### Decision · Program_Chairs · 2024-01-16

Accept (poster)